# A Memory Efficient Randomized Subspace Optimization Method for Training Large Language Models

**Yiming Chen** [*1]  **Yuan Zhang** [*2]  **Yin Liu** [1]  **Kun Yuan** [3]  **Zaiwen Wen** [1]

## Abstract

The memory challenges associated with training Large Language Models (LLMs) have become a critical concern, particularly when using the Adam optimizer. To address this issue, numerous memory-efficient techniques have been proposed, with GaLore standing out as a notable example designed to reduce the memory footprint of optimizer states. However, these approaches do not alleviate the memory burden imposed by activations, rendering them unsuitable for scenarios involving long context sequences or large minibatches. Moreover, their convergence properties are still not well-understood in the literature. In this work, we introduce a Randomized Subspace Optimization framework for pre-training and finetuning LLMs. Our approach decomposes the high-dimensional training problem into a series of lower-dimensional subproblems. At each iteration, a random subspace is selected, and the parameters within that subspace are optimized. This structured reduction in dimensionality allows our method to simultaneously reduce memory usage for both activations and optimizer states. We establish comprehensive convergence guarantees and derive rates for various scenarios, accommodating different optimization strategies to solve the subproblems. Extensive experiments validate the superior memory and communication efficiency of our method, achieving performance comparable to GaLore and Adam.

---

[*]Equal contribution  [1]Beijing International Center for Mathematical Research, Peking University, Beijing, China [2]Center for Data Science, Peking University, Beijing, China [3]Center for Machine Learning Research, Peking University, Beijing, China. Correspondence to: Kun Yuan <kunyuan@pku.edu.cn>.

*Proceedings of the $42^{nd}$ International Conference on Machine Learning*, Vancouver, Canada. PMLR 267, 2025. Copyright 2025 by the author(s).

## 1. Introduction

Large Language Models (LLMs) have achieved remarkable success across various domains (Achiam et al., 2023; Brown, 2020; Dubey et al., 2024), primarily driven by the increasing scale of datasets and model parameters. The Adam optimizer (Kingma, 2014; Loshchilov & Hutter, 2019) is widely recognized as the default choice for training these models, owing to its operation efficiency and robust performance.

However, as the scale of LLMs continues to grow, the associated memory demands have emerged as a significant bottleneck. This challenge stems from the need to store optimizer states, such as first-order and second-order moments, alongside the activations required for gradient computations. For instance, training a LLaMA-7B model necessitates 28GB of memory to store optimizer states in FP16 precision (Zhao et al., 2024a), while a GPT-3 model with 175B parameters requires an extraordinary 1.4TB memory in FP32 precision. Additionally, in scenarios involving long sequence lengths or large mini-batches, activation memory dominates as the primary constraint (Zhang et al., 2024b). These substantial memory requirements necessitate either deploying additional GPUs or reducing batch sizes. However, increasing the number of GPUs introduces additional communication overhead, potentially limiting training scalability (Malladi et al., 2023), while smaller batch sizes prolong training time due to reduced throughput.

**Memory-efficient training algorithms.** Significant efforts have been made to address the memory overhead in LLMs training. One line of research focuses on parameter-efficient methods, such as Low-Rank Adaptation (LoRA) and its variants (Hu et al., 2022; Lialin et al., 2023; Xia et al., 2024), which constrain trainable parameters to low-rank subspaces for each weight matrix. Similarly, sparsity-based techniques (Thangarasa et al., 2023) reduce memory usage by training only a subset of weights. These strategies decrease the number of trainable parameters, thereby reducing the memory requirements for storing gradients and optimizer states. Another research direction aims to achieve memory savings through the compression of optimizer states. For instance, GaLore and its variants (Chen et al., 2024b; Hao et al., 2024; He et al., 2024; Zhao et al., 2024a) project gradients onto low-rank subspaces, leveraging the compressed gradients

to compute the first- and second-order moments, which significantly reduces their memory footprint. Alternatively, Adam-mini (Zhang et al., 2024a) uses block-wise second-order moments for learning rate adjustments to reduce memory redundancy. A recent study, Apollo (Zhu et al., 2024), reinterprets Adam as an adaptive learning rate algorithm applied to the gradient. Instead of the coordinate-wise approach used in Adam, it employs a column-wise adaptive learning rate, thereby effectively reducing the memory overhead associated with optimizer states.

**Limitations in existing approaches.** Despite the progress in memory-efficient algorithms for training LLMs, two critical limitations persist in the aforementioned approaches:

**L1. Inability to reduce activations.** While the aforementioned approaches effectively reduce memory associated with optimizer states, they fail to address the memory burden posed by activations. This limitation stems from their reliance on computing full-rank gradients, which necessitates storing the complete activations. As a result, these methods are unsuitable for scenarios involving long context sequences or large mini-batches.

**L2. Insufficient convergence guarantees.** While the aforementioned approaches demonstrate strong empirical performance, their theoretical convergence properties remain less understood. For instance, GaLore (Zhao et al., 2024a) provides convergence analysis only for fixed projection matrices, rather than for the periodically updated projection matrices used in practical implementations. This lack of comprehensive theoretical guarantees raises concerns about whether these methods reliably converge to the desired solution and the rates at which such convergence occurs.

**Main results and contributions.** In this work, we propose a method that concurrently reduces memory consumption for both the optimizer states and activations. The central idea behind our approach is to decompose the original high-dimensional training problem into a series of lower-dimensional subproblems. Specifically, at each iteration, we randomly select a subspace and optimize the parameters within this subspace. After completing the optimization in one subspace, we switch to a different subspace and continue the process. Since each subproblem operates in a lower-dimensional space, it requires smaller gradients and optimizer states. As we will demonstrate, the reduced dimensionality of the subproblems also leads to a significant reduction in the memory required for storing activations. Furthermore, the smaller scale of the subproblems results in reduced communication overhead when training across multiple workers. Our main contributions are as follows:

**C1. Subspace method for LLM training.** We introduce a **R**andomized **S**ubspace **O**ptimization (**RSO**) framework for LLM training, which decomposes the original training problem into a series of lower-dimensional subproblems. This decomposition simultaneously reduces the memory required for optimizer states and activations, effectively addressing **Limitation L1**. Furthermore, the framework can reduce communication overhead in distributed training scenarios.

**C2. Theoretical convergence guarantees.** We provide a comprehensive convergence analysis for the RSO framework. The established guarantees and rates apply across various scenarios. These include subproblems solved using zeroth-order, first-order, or second-order algorithms, as well as optimization methods like gradient descent, momentum gradient descent, adaptive gradient descent, and their stochastic variants. This addresses **Limitation L2**. Notably, we present refined convergence guarantees for scenarios where subproblems are solved using the Adam optimizer.

**C3. Improved experimental performances.** We conduct extensive experiments to evaluate the proposed RSO framework. The experimental results demonstrate that our approach significantly enhances memory efficiency compared to state-of-the-art methods, such as GaLore and LoRA. Additionally, our method achieves faster training speeds by reducing communication overhead, outperforming both GaLore and Adam while maintaining comparable performance levels. These findings highlight the practical values of our approach.

## 2. Related Works

**Parameter-efficient methods.** A promising approach to memory-efficient training involves parameter-efficient methods, which reduce the number of trainable parameters and consequently lower the memory required for storing optimizer states. For example, (Hu et al., 2022) propose Low-Rank Adaptation (LoRA), which restricts trainable parameters to a low-rank subspace for each weight matrix. Similarly, (Thangarasa et al., 2023) incorporate sparsity by training only a subset of weights. While these methods effectively reduce memory consumption, the reduction in trainable parameters can sometimes lead to suboptimal model performance (Biderman et al., 2024). To address this limitation, recent advancements suggest using multiple LoRA updates to enable high-rank weight updates (Lialin et al., 2023; Xia et al., 2024). However, in pre-training settings, this approach still relies on a full-rank weight training phase as a warm-up before transitioning to low-rank training (Lialin et al., 2023), thereby limiting its memory efficiency.

**Optimizer-efficient methods.** An alternative approach to memory savings focuses on compressing optimizer states

while maintaining the number of trainable parameters. Ga-Lore (Zhao et al., 2024a) achieves this by compressing the gradient matrix through a projection onto a subspace and leveraging the compressed gradient to compute first- and second-order moments. This projection reduces the gradient size and is typically derived via the Singular Value Decomposition (SVD) of the true gradient (Zhao et al., 2024a). To mitigate the computational cost of SVD, alternative methods have been proposed, such as using random matrices (Hao et al., 2024; He et al., 2024) or generating the projection matrix through online Principal Component Analysis (PCA) (Liang et al., 2024). Fira (Chen et al., 2024a) and LDAdam (Robert et al., 2024) employ an error-feedback mechanism. The former combines the true gradient with the GaLore update to improve performance, while the latter explicitly accounts for both gradient and optimizer state compression. Apollo (Zhu et al., 2024) interprets Adam as an adaptive learning rate algorithm and uses compressed optimizer states directly as scaling factors for the true gradient. Additionally, Adafactor (Shazeer & Stern, 2018) discards the first-order moment and approximates the second-order moment with two low-rank matrices, while Adam-mini (Zhang et al., 2024a) proposes that block-wise second-order moments are sufficient for adjusting learning rates. (Das, 2024) integrates the GaLore method with a natural gradient optimizer to enhance performance. BAdam (Luo et al., 2024) and Block-LLM (Ramesh et al., 2024) incorporate block coordinate descent strategies into LLM training, restricting the number of parameters optimized in each epoch to reduce the memory overhead associated with optimizer states. Meanwhile, (Wen et al., 2025) applies wavelet transforms to compress gradients beyond the low-rank structures.

**Activation-efficient methods.** Although the aforementioned methods effectively reduce memory consumption for optimizer states, they do not address the memory costs associated with activations. To reduce activations, zeroth-order (ZO) algorithms have been introduced in LLM training (Malladi et al., 2023). These methods can be further improved through variance reduction techniques (Gautam et al., 2024), while (Zhao et al., 2024b) utilizes ZO approaches to approximate a natural gradient algorithm. Moreover, (Chen et al., 2024b) proposes a novel ZO framework to enhance performance. Unlike first-order (FO) methods, ZO algorithms approximate gradients by finite differences in function values, eliminating the need for explicit gradient computation. This approach bypasses backpropagation and activation storage, significantly reducing memory demands. However, due to their slower convergence rates (Berahas et al., 2022; Duchi et al., 2015; Nesterov & Spokoiny, 2017), ZO methods are primarily suitable for fine-tuning applications. Similarly, FO methods can achieve activation savings by layer-wise training (Lai et al., 2024), but their use also predominantly targets fine-tuning phases.

**System-based methods.** Several system-level techniques have been proposed to improve memory efficiency. Activation checkpointing (Chen et al., 2016) reduces memory usage by recomputing activations on demand rather than storing them throughout the entire iteration, though this comes at the cost of increased computational complexity. Quantization (Dettmers et al., 2023) lowers memory consumption by using lower-bit data representations, but this may introduce a trade-off between memory efficiency and training precision. Additionally, methods such as those introduced by (Ren et al., 2021; Zhang et al., 2023a) reduce GPU memory usage by offloading data to non-GPU resources, which can lead to additional communication overhead.

# 3. Preliminaries

This section introduces the optimization framework for LLM pre-training and fine-tuning, followed by a review of several memory-efficient methods.

## 3.1. LLM Optimization

When addressing the pre-training or fine-tuning of LLMs, the problem can be formulated as follows:

$$\min_{\boldsymbol{W}} f(\boldsymbol{W}) := \mathbb{E}_\xi \left[ F(\boldsymbol{W}; \xi) \right], \qquad (1)$$

where $\boldsymbol{W} = \{W_\ell\}_{\ell=1}^{\mathcal{L}}$ represents the set of trainable parameters with a total dimension of $d$. Here, $W_\ell \in \mathbb{R}^{m_\ell \times n_\ell}$ denotes the weight matrix for the $\ell$-th layer, and $\mathcal{L}$ is the total number of layers. The function $F(\boldsymbol{W}; \xi)$ is the loss function, which depends on the random variable $\xi$ representing individual data samples.

To address the optimization problem defined in (1), commonly used approaches include SGD (Bottou, 2010), Momentum SGD (Sutskever et al., 2013), and Adam (Kingma, 2014). The iterative update rule for Adam is as follows:

$$\mathbf{M}^t = \beta_1 \cdot \mathbf{M}^{t-1} + (1 - \beta_1) \cdot \nabla F(\boldsymbol{W}^t; \xi^t), \qquad (2a)$$

$$\mathbf{V}^t = \beta_2 \cdot \mathbf{V}^{t-1} + (1 - \beta_2) \cdot (\nabla F(\boldsymbol{W}^t; \xi^t))^2, \qquad (2b)$$

$$\hat{\mathbf{M}}^t = \mathbf{M}^t / (1 - \beta_1^t), \quad \hat{\mathbf{V}}_t = \mathbf{V}_t / (1 - \beta_2^t), \qquad (2c)$$

$$\boldsymbol{W}^{t+1} = \boldsymbol{W}^t - \alpha \cdot \hat{\mathbf{M}}^t / (\sqrt{\hat{\mathbf{V}}^t} + \epsilon). \qquad (2d)$$

Here, $\mathbf{M}$ and $\mathbf{V}$ represent the first-order and second-order moments, respectively, and $\epsilon > 0$ is a small constant.

## 3.2. Memory Consumption in LLM Training

The key memory components involved in the training process include four primary elements: model parameters, optimizer states, gradients, and activations. The model component stores parameters required for training. In the case of the Adam optimizer, the optimizer states are represented by the first and second moment estimates, denoted as $\mathbf{M}$ and

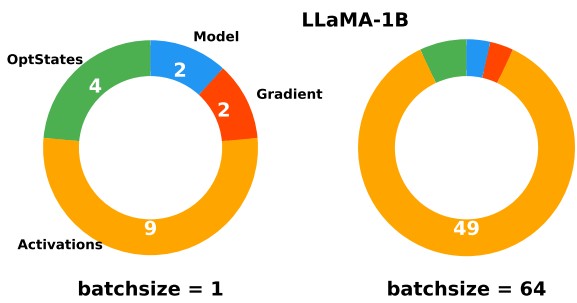

*Figure 1.* Memory components involved in training the LLaMA-1B model using the Adam optimizer under varying batch sizes. The reported values indicate memory usage in GB.

$\boldsymbol{V}$. The gradient corresponds to the memory cost associated with $\nabla F(\boldsymbol{W}; \xi)$. With the Adam optimizer, both the optimizer state and the gradient are determined by the number of trainable parameters, see recursions (2a)–(2b).

Another significant memory cost arises from the activations, which represent the intermediate values computed during forward propagation. Unlike the optimizer state and gradients, the memory for activations depends on multiple factors, including model size, batch size, and sequence length.

Figure 1 illustrates the memory consumption during the training of the LLaMA-1B model. For small batch sizes, the optimizer state constitutes a substantial portion of the memory usage. In contrast, for large batch sizes, activations dominate and account for nearly the entire memory cost.

### 3.3. Memory-efficient Method

As previously discussed, the optimizer state imposes a substantial memory overhead. To address this challenge, Ga-Lore (Zhang et al., 2023b) introduces a projection technique that generates a compressed representation of the optimizer state, eliminating the need to store its full version. Consequently, the update rule of GaLore is as follows:

$$\tilde{\mathbf{M}}^t = \beta_1 \cdot \tilde{\mathbf{M}}^{t-1} + (1-\beta_1) \cdot \boldsymbol{P}^\top \nabla F(\boldsymbol{W}^t; \xi^t), \quad (3a)$$

$$\tilde{\mathbf{V}}^t = \beta_2 \cdot \tilde{\mathbf{V}}^{t-1} + (1-\beta_2) \cdot (\boldsymbol{P}^\top \nabla F(\boldsymbol{W}^t; \xi^t))^2, \quad (3b)$$

$$\hat{\mathbf{M}}_t = \tilde{\mathbf{M}}_t/(1-\beta_1^t), \quad \hat{\mathbf{V}}_t = \tilde{\mathbf{V}}_t/(1-\beta_2^t), \quad (3c)$$

$$\boldsymbol{W}^{t+1} = \boldsymbol{W}^t - \alpha \cdot \boldsymbol{P} \, \hat{\mathbf{M}}_t/(\sqrt{\hat{\mathbf{V}}_t} + \epsilon). \quad (3d)$$

Here, $\boldsymbol{P}$ represents the projection matrix, which maps the gradient matrix onto a lower-dimensional subspace. Specifically, GaLore selects $\boldsymbol{P}$ as the top left singular vectors of the gradient matrix, capturing its most important components.

Since the projected gradient $\boldsymbol{P}^\top \nabla F(\boldsymbol{W}^t; \xi^t)$ lies within a low-dimensional subspace, the associated optimizer states $\tilde{\mathbf{M}}$ and $\tilde{\mathbf{V}}$ in GaLore are also substantially reduced in size.

This leads to notable memory savings compared to the Adam optimizer. However, as shown in Figure 1, the optimizer state contributes significantly to memory costs primarily when using a small batch size. Conversely, with larger batch sizes—more practical in many scenarios—the memory efficiency advantages of GaLore diminish, as activation memory becomes the dominant component of overall memory consumption.

## 4. Randomized Subspace Optimization

In this section, we present the randomized subspace optimization (RSO) method, tailored explicitly for the pretraining and fine-tuning of LLMs.

### 4.1. Algorithm Framework

As previously discussed, the memory overhead in LLM training primarily stems from the large scale of the models. In other words, the primary source of memory consumption arises from the high dimensionality of the LLM training problem (1). This observation motivates us to decompose the original problem into a series of lower-dimensional subproblems. By partitioning the problem into smaller components, we can effectively reduce memory usage, as each subproblem requires less memory to process.

Similar to the random coordinate descent (Wright, 2015), which optimizes the objective function one coordinate at a time, we address problem (1) incrementally, subspace by subspace. The proposed update rules are as follows:

$$\tilde{\boldsymbol{B}}^k \approx \arg\min_{\boldsymbol{B}} \left\{ f(\boldsymbol{W}^k + \boldsymbol{P}^k \boldsymbol{B}) + \frac{1}{2\eta^k} \|\boldsymbol{B}\|^2 \right\}, \quad (4a)$$

$$\boldsymbol{W}^{k+1} = \boldsymbol{W}^k + \boldsymbol{P}^k \tilde{\boldsymbol{B}}^k, \quad (4b)$$

Here, $f(\boldsymbol{W}^k + \boldsymbol{P}^k \boldsymbol{B}) = \mathbb{E}_\xi[F(\boldsymbol{W}^k + \boldsymbol{P}^k \boldsymbol{B}; \xi)]$ in which $\boldsymbol{P}^k = \{P_\ell^k\}_{\ell=1}^{\mathcal{L}}$ denotes the subspace projection matrices. Each $P_\ell^k \in \mathbb{R}^{m_\ell \times r_\ell}$ is a randomly selected matrix with $r_\ell \ll m_\ell$. The parameters $\boldsymbol{B} = \{B_\ell\}_{\ell=1}^{\mathcal{L}}$ consist of variables with significantly smaller dimensions compared to $\boldsymbol{W}$. Specifically, in the $\ell$-th layer, $B_\ell$ has dimensions $r_\ell \times n_\ell$, whereas $W_\ell$ has dimensions $m_\ell \times n_\ell$. A proximal term $\|\boldsymbol{B}\|^2 := \sum_\ell \|B_\ell\|_F^2$ is introduced to (4a) to ensure convergence, with coefficient $\eta^k$ to regulate its influence.

In the $k$-th iteration, a subspace projection matrix $\boldsymbol{P}^k$ is randomly selected, and the subproblem in (4a) is solved. This process approximately minimizes the objective function within the chosen subspace. Upon solving the subproblem, the current parameters are updated, and a new subspace projection matrix, $\boldsymbol{P}^{k+1}$, is selected for the subsequent iteration. When addressing the subproblem in (4a), standard optimizers such as GD, SGD, momentum SGD or Adam can be employed. Notably, obtaining an exact solution in (4a) is not required; an inexact solution suffices for the proposed

approach. The RSO algorithm is presented in Algorithm 1.

---

**Algorithm 1** Randomized Subspace Optimization

---

**Input:** Initialization $\boldsymbol{W}^0$.
**Output:** Solution $\boldsymbol{W}^K$.

1: **for** $k = 0, 1, \ldots, K - 1$ **do**
2:     Sample $\boldsymbol{P}^k$ according to a given distribution.
3:     Solve subproblem (4a) and obtain the approximate solution $\tilde{\boldsymbol{B}}^k$ using a given optimizer such as Adam.
4:     Update the weights by $\boldsymbol{W}^{k+1} = \boldsymbol{W}^k + \boldsymbol{P}^k \tilde{\boldsymbol{B}}^k$.
5: **end for**

---

### 4.2. Memory Efficiency

We now demonstrate that the proposed RSO approach offers superior memory efficiency. Unlike other memory-efficient methods (e.g., GaLore, Adam-mini, Apollo, etc.) that primarily focus on reducing the memory usage of optimizer states, the RSO method additionally achieves substantial savings in gradient and activation memory requirements.

**Memory for optimizer states.** When solving (4a), the reduced dimensionality of the subproblem significantly decreases the memory requirements for optimizer states. For instance, the memory required for both the first-order and second-order moment estimates in each subproblem is $r_\ell n_\ell$ parameters per $\ell$-th layer, which is substantially lower than the $m_\ell n_\ell$ memory overhead in the standard Adam optimizer.

**Memory for gradients.** Specifically, for the subproblems in (4a), it is sufficient to compute the gradient with respect to $\boldsymbol{B}$, i.e., $\nabla_{\boldsymbol{B}} F(\boldsymbol{W}^k + \boldsymbol{P}^k \boldsymbol{B}; \xi)$, rather than calculating the full-dimensional gradient $\nabla_{\boldsymbol{W}} F(\boldsymbol{W}^k; \xi)$ with respect to the original weight matrix $\boldsymbol{W}$, as outlined in the GaLore recursion in (3a)–(3b). This results in considerable memory savings associated with the gradient computation.

**Memory for activations.** The RSO method not only reduces memory usage for gradients but also significantly minimizes the memory required to store activations. For example, consider a neural network where the $\ell$-th layer is defined as follows:

$$(\text{Adam}): \quad Z_\ell = Y_\ell \cdot W_\ell, \qquad y = L(Z_\ell). \quad (5)$$
$$(\text{RSO}): \quad Z_\ell = Y_\ell \cdot (W_\ell + P_\ell B_\ell), \quad y = L(Z_\ell). \quad (6)$$

Expression (5) represents the forward process of Adam, where the $\ell$-th layer is associated with the weight matrix $W_\ell$, while (6) corresponds to the RSO method associated with the weight matrix $B_\ell$. Here, $Y_\ell \in \mathbb{R}^{s_\ell \times m_\ell}$ denotes the output of the previous layer (i.e., the activation), and $Z_\ell$ serves as the input to the next layer. The function $L(\cdot)$, encompassing all subsequent layers and the loss function, depends only on $Z_\ell$ and not on $Y_\ell$. Thus, once $Z_\ell$ is computed, $Y_\ell$ is no longer required for calculating the loss $y$.

| Algorithm | Memory | |
|---|---|---|
| | Optimizer States | Activations |
| **RSO** | $24nr$ | $8bsn + 4bsr + 2bs^2$ |
| GaLore | $24nr$ | $15bsn + 2bs^2$ |
| LoRA | $48nr$ | $15bsn + 2bs^2$ |
| Adam | $24n^2$ | $15bsn + 2bs^2$ |

*Table 1.* Memory analysis of different algorithms in terms of optimizer states and activations for one typical transformer block. Here, $s$, $b$, and $n$ represent the sequence length, batch size, and embedding dimension, respectively. The intermediate dimension of the feed-forward network is assumed to be $4n$.

In the backward-propagation process, Adam and RSO computes the weight gradient as follows:

$$(\text{Adam}): \quad \frac{\partial y}{\partial W_\ell} = Y_\ell^\top \frac{\partial y}{\partial Z_\ell}, \quad (7)$$

$$(\text{RSO}): \quad \frac{\partial y}{\partial B_\ell} = (Y_\ell P_\ell)^\top \frac{\partial y}{\partial Z_\ell}. \quad (8)$$

Adam requires storing the activation $Y_\ell \in \mathbb{R}^{s_\ell \times m_\ell}$ in (7) to compute gradients with respect to $W_\ell$. In contrast, RSO only needs to store $Y_\ell P_\ell \in \mathbb{R}^{s_\ell \times r_\ell}$ to compute gradients with respect to $B_\ell$. Since $r_\ell \ll m_\ell$, this approach achieves significant memory savings. As a result, the RSO method substantially reduces the memory overhead associated with activations in layers of the form (6).

We analyze the memory overhead of the proposed RSO method for a typical transformer block. Table 1 summarizes the results, comparing memory usage for optimizer states and activations across various algorithms. Details of this memory analysis is presented in Appendix A.

### 4.3. Communication Efficiency

As previously mentioned, the RSO algorithm solves smaller subproblems at each iteration, resulting in gradients with reduced dimensionality compared to methods like Adam and GaLore, which rely on full-dimensional gradients. This reduction in gradient size enables RSO to achieve improved communication efficiency.

Specifically, in a data-parallel framework such as Distributed Data Parallel (DDP), the model is replicated across multiple devices, with each device computing gradients on its local data batch. These gradients are then aggregated across devices, necessitating gradient communication. By operating with lower-dimensional gradients, the RSO method effectively reduces communication overhead compared to existing approaches.

| Subproblem Solver | Subproblem Complexity | Total Complexity |
|---|---|---|
| Zero-Order (ZO) Methods | | |
| Stochastic ZO Method (Shamir, 2013) | $O\big((\sum_{\ell=1}^{\mathcal{L}} n_\ell r_\ell)^2 \epsilon^{-2}\big)$ | $O\big((\sum_{\ell=1}^{\mathcal{L}} n_\ell r_\ell)^2 \epsilon^{-3}\big)$ |
| First-Order (FO) Methods | | |
| GD | $O(\log \epsilon^{-1})$ | $\tilde{O}(\epsilon^{-1})$ |
| Accelerated GD (Nesterov et al., 2018) | $O(\log \epsilon^{-1})$ | $\tilde{O}(\epsilon^{-1})$ |
| SGD (Bottou et al., 2018) | $O(\epsilon^{-1})$ | $O(\epsilon^{-2})$ |
| Momentum SGD (Yuan et al., 2016) | $O(\epsilon^{-1})$ | $O(\epsilon^{-2})$ |
| Adam-family (Guo et al., 2024) | $O(\epsilon^{-1})$ | $O(\epsilon^{-2})$ |
| Second-Order (SO) Methods | | |
| Newton's method (Boyd & Vandenberghe, 2004) | $O(\log(\log \epsilon^{-1}))$ | $\tilde{O}(\epsilon^{-1})$ |
| Stochastic Quasi-Newton method (Byrd et al., 2016) | $O(\epsilon^{-1})$ | $O(\epsilon^{-2})$ |

*Table 2.* The sample complexities of the RSO method with various subproblem solvers. For the ZO solver, it refers to the number of stochastic function value evaluations; for the FO solver, it refers to the number of deterministic/stochastic gradient computations; and for the SO solver, it refers to the number of deterministic/stochastic Hessian or estimated Hessian computations. $\tilde{O}(\cdot)$ hides logarithm terms.

## 5. Convergence Analysis

In this section, we present the convergence guarantees for the RSO method. To account for the use of various optimizers in solving the subproblem (4a), we assume that, at each iteration $k$, the chosen optimizer produces an expected $\epsilon$-inexact solution. Such an expected $\epsilon$-inexact solution is defined below:

**Definition 5.1** (Expected $\epsilon$-inexact solution)**.** *A solution $\tilde{\boldsymbol{B}}^k$ is said to be an expected $\epsilon$-inexact solution if it satisfies:*

$$\mathbb{E}[g^k(\tilde{\boldsymbol{B}}^k)] - g^k(\boldsymbol{B}_\star^k) \leq \epsilon, \qquad (9)$$

*where $g^k(\boldsymbol{B}) := f(\boldsymbol{W}^k + \boldsymbol{P}^k \boldsymbol{B}) + \frac{1}{2\eta^k}\|\boldsymbol{B}\|^2$, and $\boldsymbol{B}_\star^k$ is the optimal solution define as $\boldsymbol{B}_\star^k := \arg\min_{\boldsymbol{B}} g^k(\boldsymbol{B})$.*

When $\eta^k$ is properly chosen, it can be guaranteed that $g^k(\boldsymbol{B})$ is a strongly convex function hence $\boldsymbol{B}_\star^k$ is unique.

To establish convergence guarantees for the RSO algorithm, we require the following assumptions:

**Assumption 5.2.** *The objective function $f(\boldsymbol{W})$ is $L$-smooth, i.e., it holds for any $\boldsymbol{W}^1$ and $\boldsymbol{W}^2$ that*

$$\|\nabla f(\boldsymbol{W}^1) - \nabla f(\boldsymbol{W}^2)\| \leq L\|\boldsymbol{W}^1 - \boldsymbol{W}^2\|,$$

*where $\|\boldsymbol{W}\| := \sqrt{\sum_{\ell=1}^{\mathcal{L}} \|W_\ell\|_F^2}$ for any $\boldsymbol{W} = \{W_\ell\}_{\ell=1}^{\mathcal{L}}$.*

**Assumption 5.3.** *The random matrix $\boldsymbol{P} = \{P_\ell\}_{\ell=1}^{\mathcal{L}}$ is sampled from a distribution such that $P_\ell^\top P_\ell = (m_\ell/r_\ell)I_{r_\ell}$ and $\mathbb{E}[P_\ell P_\ell^\top] = I_{m_\ell}$ for each $\ell$.*

**Remark 5.4.** *In practice, when $m_\ell \gg r_\ell$, sampling each $P_\ell$ from a normal distribution $\mathcal{N}(0, \frac{1}{r_\ell})$ yields an approximation $P_\ell^\top P_\ell \approx (m_\ell/r_\ell)I_{r_\ell}$. This approach provides computational efficiency. However, to rigorously satisfy Assumption*

*5.3, $P_\ell$ should be drawn from a Haar distribution or constructed as a random coordinate matrix (see (Kozak et al., 2023), Examples 1 and 2 for further details).*

The following theorem establish the convergence rate of the RSO algorithm. Detailed proofs are provided in Appendix B.

**Theorem 5.5.** *Under Assumptions 5.2 and 5.3, let each subproblem in (4b) be solved starting from the initial point $\boldsymbol{B}^0 = \boldsymbol{0}$ to an expected $\epsilon$-inexact solution $\tilde{\boldsymbol{B}}^k$ with suitable choice of $\eta^k$. The sequence $\{\boldsymbol{W}^k\}$ generated by the RSO method satisfies the following bound:*

$$\frac{1}{K}\sum_{k=0}^{K-1} \mathbb{E}\|\nabla f(\boldsymbol{W}^k)\|^2 \leq \frac{18\hat{L}\Delta_0}{K} + 18\hat{L}\epsilon, \qquad (10)$$

*where $\Delta_0 := f(\boldsymbol{W}^0) - f^*$ and $\hat{L} := \max_\ell\{m_\ell/r_\ell\}L$.*

**Sample complexity with different optimizers.** When all subproblems are solved to expected $\epsilon$-inexact solutions, the RSO method achieves an $\epsilon$-stationary point within $O(\epsilon^{-1})$ iterations. As each iteration requires solving the subproblem (4a), the total sample complexity of the RSO method depends on the solver employed for this subproblem. For instance, since gradient descent solves (4a) in $O(\log \epsilon^{-1})$ inner iterations, the RSO method using gradient descent attains a total sample complexity of $O(\epsilon^{-1} \log \epsilon^{-1})$. Table 2 summarizes the sample complexities for the RSO method when equipped with various solvers, including zeroth-order, first-order, and second-order scenarios with optimizers such as gradient descent, momentum gradient descent, adaptive gradient descent, and their stochastic variants.

**Comparable complexity with vanilla Adam.** It is observed

| Algorithm | 60M | 130M | 350M | 1B |
|---|---|---|---|---|
| Adam* | 34.06 (0.22G) | 25.08 (0.50G) | 18.80 (1.37G) | 15.56 (4.99G) |
| GaLore* | 34.88 (0.14G) | 25.36 (0.27G) | 18.95 (0.49G) | **15.64** (1.46G) |
| LoRA* | 34.99 (0.16G) | 33.92 (0.35G) | 25.58 (0.69G) | 19.21 (2.27G) |
| ReLoRA* | 37.04 (0.16G) | 29.37 (0.35G) | 29.08 (0.69G) | 18.33 (2.27G) |
| **RSO** | **34.55**(0.14G) | **25.34** (0.27G) | **18.86** (0.49G) | 15.86 (1.46G) |
| $r/d_{model}$ | 128 / 256 | 256 / 768 | 256 / 1024 | 512 / 2048 |
| Training Tokens (B) | 1.1 | 2.2 | 6.4 | 13.1 |

*Table 3.* Comparison of validation perplexity and estimated memory usage for optimizer states across different algorithms during the pre-training of LLaMA models of various sizes on the C4 dataset. The optimizer states are stored in BF16 format. Results marked with * are sourced from (Zhao et al., 2024a).

| | CoLA | STS-B | MRPC | RTE | SST2 | MNLI | QNLI | QQP | Avg |
|---|---|---|---|---|---|---|---|---|---|
| Adam | 62.24 | 90.92 | 91.30 | 79.42 | 94.57 | 87.18 | 92.33 | 92.28 | 86.28 |
| GaLore (rank=4) | 60.35 | **90.73** | **92.25** | **79.42** | 94.04 | **87.00** | 92.24 | 91.06 | 85.89 |
| LoRA (rank=4) | 61.38 | 90.57 | 91.07 | 78.70 | 92.89 | 86.82 | 92.18 | **91.29** | 85.61 |
| **RSO** (rank=4) | **62.47** | 90.62 | **92.25** | 78.70 | **94.84** | 86.67 | **92.29** | 90.94 | **86.10** |
| GaLore (rank=8) | 60.06 | **90.82** | 92.01 | **79.78** | 94.38 | **87.17** | 92.20 | 91.11 | 85.94 |
| LoRA (rank=8) | 61.83 | 90.80 | 91.90 | 79.06 | 93.46 | 86.94 | 92.25 | 91.22 | 85.93 |
| **RSO** (rank=8) | **64.62** | 90.71 | **93.56** | 79.42 | **95.18** | 86.96 | **92.44** | 91.26 | **86.77** |

*Table 4.* Evaluation of various fine-tuning methods on the GLUE benchmark using the pre-trained RoBERTa-Base model. The average score across all tasks is provided.

in Table 2 that RSO with Adam to solve subproblem has sample complexity $O(\epsilon^{-2})$, which is on the same order as vanilla Adam (Kingma, 2014) without subspace projection.

# 6. Experiments

In this section, we present numerical experiments to evaluate the effectiveness of our RSO method. We assess its performance on both pre-training and fine-tuning tasks across models of varying scales. Additionally, we compare the memory usage and time cost of our RSO method with existing approaches to highlight its advantages in memory and communication efficiency. In all tests, the RSO method uses the Adam optimizer with a fixed number of steps to solve each subproblem. The random projection matrices $\boldsymbol{P} = \{P_\ell\}_{\ell=1}^{\mathcal{L}}$ are independently sampled from a normal distribution $\mathcal{N}(0, 1/r_\ell)$ for each layer. We set all $r_\ell$ to the same value, which we define as the "rank" of our method to facilitate consistent comparison with other approaches that explicitly specify a rank parameter.

## 6.1. Pre-training with RSO

**Experimental setup.** We evaluate the performance of our RSO method on LLaMA models with sizes ranging from 60M to 7B parameters. The experiments are conducted

using the C4 dataset, a large-scale, cleaned version of Common Crawl's web corpus, which is primarily intended for pre-training language models and word representations (Raffel et al., 2020). We compare our method against LoRA (Hu et al., 2022), GaLore (Zhao et al., 2024a), ReLoRA (Lialin et al., 2023), and Adam (Kingma, 2014) as baseline methods. We adopt the same configurations as those reported in (Zhao et al., 2024a), and the detailed settings for pre-training are provided in Appendix D.1.

**Main results.** As shown in Table 3, under the same rank constraints, RSO outperforms other memory-efficient methods in most cases. We also report the estimated memory overhead associated with optimizer states. From Table 3, we observe that for LLaMA-350M, RSO achieves nearly the same performance as Adam while reducing the memory required for optimizer states by 64.2%. For LLaMA-1B, this reduction increases to 70.7%. The results for LLaMA-7B are provided in Appendix C.1.

## 6.2. Fine-tuning with RSO

**Experimental setup.** We extend the application of our RSO algorithm to fine-tuning tasks. Specifically, we fine-tune pre-trained RoBERTa models (Liu et al., 2019) on the GLUE benchmark (Wang et al., 2019), which encompasses a diverse range of tasks, including question answering, sen-

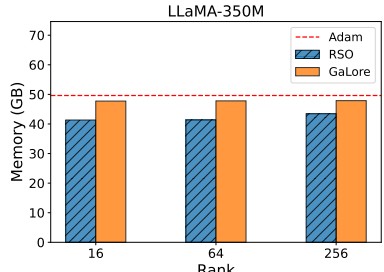 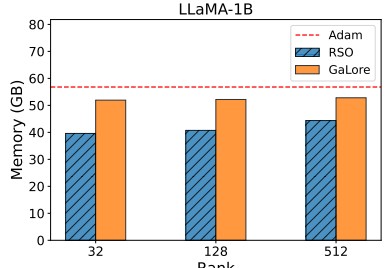 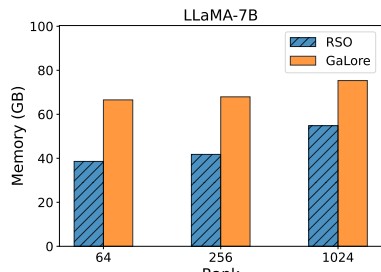

*Figure 2.* Comparison of peak memory usage (in GB) per device for RSO and GaLore during LLaMA training with varying ranks. All hyperparameters, except rank, are consistent with (Zhao et al., 2024a). Adam's memory usage is reported for LLaMA-350M and LLaMA-1B but excluded for LLaMA-7B due to an out-of-memory (OOM) error.

| Method | LLaMA-1B (Seconds) | | | LLaMA-7B (Seconds) | | |
|--------|:------:|:-------:|:-------:|:------:|:-------:|:-------:|
|        | Seq 64 | Seq 128 | Seq 256 | Seq 64 | Seq 128 | Seq 256 |
| RSO    | 0.94 | 1.70 | 3.29 | 2.40 | 2.94 | 4.60 |
| GaLore | 1.12 | 1.84 | 3.35 | 7.86 | 8.26 | 9.12 |
| Adam   | 1.11 | 1.81 | 3.32 | 7.84 | 8.23 | OOM  |

*Table 5.* Comparison of iteration time (in seconds) for different methods in LLaMA training across various sequence lengths. All hyperparameters, except sequence length, follow (Zhao et al., 2024a). LLaMA-1B runs on $4\times$ A800 GPUs, while LLaMA-7B uses $8\times$ A800 GPUs. SVD decomposition time in GaLore is excluded. Additionally, for LLaMA-7B with a sequence length of 256, the Adam optimizer encounters an out-of-memory (OOM) error.

timent analysis, and semantic textual similarity. Detailed settings can be found in Appendix D.2.

**Main results.** As shown in Table 4, our RSO method surpasses other memory-efficient approaches and delivers performance comparable to Adam across most datasets in the GLUE benchmark. Notably, RSO significantly outperforms Adam on the CoLA and MRPC datasets when the rank is set to 8. Additional fine-tuning experiments on LLaMA and OPT models are provided in Appendix C.2.

### 6.3. Memory and Communication Efficiency

To evaluate the memory and communication efficiency of our proposed method, we measure the peak memory usage and iteration time during the pre-training of LLaMA models of various sizes.

**RSO method requires less memory overhead.** Figure 2 illustrates the actual memory usage during the training of LLaMA models. As shown, the RSO method incurs significantly lower memory overhead compared to GaLore and Adam. This reduction is attributed to RSO's ability to save memory for activations and use low-dimensional gradients. For instance, in the case of LLaMA-7B with a rank of 64, the RSO method achieves over a 40% reduction in memory overhead compared to GaLore.

Additionally, in Figure 2, the memory gap between RSO and

GaLore widens as the rank decreases. This is because, as indicated in Table 1, the RSO method further reduces memory consumption for activations with lower ranks, whereas GaLore does not benefit in this regard. For LLaMA-350M and LLaMA-1B, GaLore's memory usage is observed to be comparable to that of Adam, as activation memory dominates in these cases. However, RSO still achieves superior memory efficiency due to its reduced activation cost.

**RSO method requires less time per iteration.** Table 5 presents a comparison of the time required for one iteration across different methods when training LLaMA models. As shown, the RSO method requires significantly less time compared to GaLore or Adam due to its improved communication efficiency, achieved by reducing the dimensionality of gradients. For example, when training LLaMA-7B with a sequence length of 64, the time required by RSO is only one-third that of GaLore or Adam. Notably, while GaLore involves SVD decomposition (which is excluded from this measurement), RSO demonstrates even greater efficiency in training time.

As shown in Table 5, the difference in iteration time between RSO and other approaches becomes more pronounced as model size increases or sequence length decreases. This phenomenon can be attributed to the communication overhead, which primarily stems from the synchronization of gradients across devices. Such overhead is more closely tied to model size while being less affected by sequence length.

In contrast, the computational overhead is highly sensitive to sequence length. Therefore, RSO exhibits a more substantial advantage when the sequence length is considerably smaller than the model size, as communication overhead constitutes a larger fraction of the total iteration time under these conditions.

## 7. Conclusion

We propose a Randomized Subspace Optimization (RSO) method, aiming for Large Language Model (LLM) pre-training and fine-tuning. By decomposing the origanal training problem into small subproblems, our method achieves both memory and communication efficiency, while reach same level performance compared with GaLore and Adam.

We outline two directions for future work on the RSO method. First, further reduction of memory overhead for activations could be explored. While part of the activation memory has already been reduced, the remaining portion might be further optimized through alternative strategies for partitioning the original problem. Second, it is worth investigating the performance of various methods for solving the subproblems. Given the low-dimensional nature of the subproblems, exploring the application of second-order methods could be particularly promising.

## Impact Statement

This study focuses on improving the memory efficiency of large language model (LLM) training. By enabling LLM training on devices with lower memory capacity, our approach helps reduce resource consumption and carbon emissions associated with LLM training. Furthermore, this enhancement may make LLM training more accessible to smaller institutions and research teams operating under constrained budgets.

## Acknowledgments

The computational resources were supported by the Center for Intelligent Computing and Song-Shan Lake HPC Center (SSL-HPC) in Great Bay University, Dongguan, China. This work was supported in part by National Key Research and Development Program of China under the grant numbers 2024YFA1012902 and 2024YFA1012903, and the National Natural Science Foundation of China under the grant numbers 12331010, 12288101, 12401408 and W2441021. Kun Yuan is also supported by AI for Science Institute, Beijing, China, and National Engineering Labratory for Big Data Analytics and Applications. We also thank the anonymous reviewers for their valuable feedback.

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

# A. Memory Complexity Analysis

In this section, we analyze the memory overhead of our proposed RSO algorithm for one typical transformer block.

## A.1. Transformer Structure

Transformers (Vaswani, 2017) have become a foundational component of LLMs. Here, we focus on the forward and backward propagation processes within a single transformer block using our RSO algorithm.

**Forward Propagation.** Consider the input $X \in \mathbb{R}^{s \times n}$ to a transformer block, where $s$ is the sequence length and $n$ is the embedding dimension. The attention mechanism within the transformer block performs the following linear operations:

$$Q = X(W_q + P_q B_q), \quad K = X(W_k + P_k B_k), \quad V = X(W_v + P_v B_v), \tag{11}$$

where $W_q, W_k, W_v \in \mathbb{R}^{n \times n}$ are the original weight matrices, $P_q, P_k, P_v \in \mathbb{R}^{n \times r}$ are the projection matrices, and $B_q, B_k, B_v \in \mathbb{R}^{r \times n}$ are the low-rank weight matrices used in the RSO method for each subproblem. These intermediate values are then combined as follows:

$$\tilde{A}_s = QK^\top, \quad A_s = \sigma_s\left(\frac{\tilde{A}_s}{\sqrt{n}}\right), \quad A_h = A_s V, \quad A_o = A_h(W_o + P_o B_o), \tag{12}$$

where $\sigma_s$ represents the softmax activation function, and $W_o \in \mathbb{R}^{n \times n}$ is the output projection matrix.

Next, the feed-forward network consists of two fully-connected layers, which are computed as:

$$\tilde{Z}_1 = A_o(W_1 + P_1 B_1), \quad Z_1 = \sigma(\tilde{Z}_1), \quad Z_2 = Z_1(W_2 + P_2 B_2), \tag{13}$$

where $W_1 \in \mathbb{R}^{n \times 4n}$ and $W_2 \in \mathbb{R}^{4n \times n}$ are the weights of the feed-forward layers. We assume that the intermediate dimension of the feed-forward network is four times the embedding dimension. Similarly, $P_1 \in \mathbb{R}^{n \times r}, P_2 \in \mathbb{R}^{4n \times r}$ are the projection matrices, and $B_1 \in \mathbb{R}^{r \times 4n}, B_2 \in \mathbb{R}^{r \times n}$ are the low-rank trainable parameters for RSO. The function $\sigma$ represents the activation function.

For the Adam and GaLore methods, there are no $P$ and $B$ matrices, as they directly work with the original weight matrices. In the case of the LoRA method, each projection matrix $P$ is replaced by a trainable parameter $A$.

**Backward Propagation.** To calculate the gradients of all the weight matrices, the backward propagation begins with the partial gradient of the loss function $F$ with respect to the output of this block, denoted as $\mathcal{D}Z_2 := \frac{\partial F}{\partial Z_2}$. Here, we use $\mathcal{D}$ to represent the derivative of $F$ with respect to any matrix. Note that in our RSO algorithm, we compute the gradient with respect to $B$, the low-rank trainable parameters, instead of the original weight matrix $W$. The gradients for the weights in the feed-forward network are computed as follows:

$$\mathcal{D}B_2 = (Z_1 P_2)^\top \mathcal{D}Z_2, \ \mathcal{D}\tilde{Z}_1 = \mathcal{D}Z_2(W_2 + P_2 B_2)^\top \odot \sigma'(\tilde{Z}_1), \ \mathcal{D}B_1 = (A_o P_1)^\top \mathcal{D}\tilde{Z}_1, \ \mathcal{D}A_o = \mathcal{D}\tilde{Z}_1(W_1 + P_1 B_1)^\top. \tag{14}$$

For the attention mechanism, the gradients for the corresponding matrices are calculated as:

$$\mathcal{D}B_o = (A_h P_o)^\top \mathcal{D}A_o, \quad \mathcal{D}A_h = \mathcal{D}A_o(W_o + P_o B_o)^\top, \quad \mathcal{D}A_s = \mathcal{D}A_h V^\top. \tag{15}$$

To compute the gradients for the matrices $Q, K, V$, the following equations are used:

$$\mathcal{D}V = A_s^\top \mathcal{D}A_h, \quad \mathcal{D}Q = \left[\mathcal{D}A_s \odot \frac{1}{\sqrt{n}}\sigma_s'\left(\frac{\tilde{A}_s}{\sqrt{n}}\right)\right] K, \quad \mathcal{D}K = \left[\mathcal{D}A_s \odot \frac{1}{\sqrt{n}}\sigma_s'\left(\frac{\tilde{A}_s}{\sqrt{n}}\right)\right]^\top Q. \tag{16}$$

The gradients for the low-rank weight matrices $B_q, B_k, B_v$ are computed as follows:

$$\mathcal{D}B_v = (XP_v)^\top \mathcal{D}V, \quad \mathcal{D}B_q = (XP_q)^\top \mathcal{D}Q, \quad \mathcal{D}B_k = (XP_k)^\top \mathcal{D}K. \tag{17}$$

Finally, to ensure that the backward propagation process can continue, the derivative with respect to the input $X$ must also be calculated. This is given by:

$$\mathcal{D}X = \mathcal{D}Q(W_q + P_q B_q)^\top + \mathcal{D}K(W_k + P_k B_k)^\top + \mathcal{D}V(W_v + P_v B_v)^\top. \tag{18}$$

When using the Adam or GaLore algorithms, the derivatives must be computed with respect to the original weight matrix $W$ instead of the low-rank matrix $B$. As a result, all occurrences of $\mathcal{D}B$ need to be replaced with $\mathcal{D}W$. For example, the derivatives with respect to $W_q, W_k$, and $W_v$ are computed as follows:

$$\mathcal{D}W_v = X^\top \mathcal{D}V, \quad \mathcal{D}W_q = X^\top \mathcal{D}Q, \quad \mathcal{D}W_k = X^\top \mathcal{D}K.$$

### A.2. Memory for Optimizer States Analysis

For Adam algorithm, the trainable parameters include $W_q, W_k, W_v, W_o, W_1, W_2$. It is straightforward to compute the total number of parameters as $12n^2$. Consequently, the optimizer states, considering both the first-order and second-order moments, require $24n^2$ storage.

For RSO method, the trainable weights for each subproblem are $B_q, B_k, B_v, B_o, B_1, B_2$, with a total of $12nr$ parameters per subproblem, leading to an optimizer state storage requirement of $24nr$. LoRA trains an additional matrix $A$ (corresponding to the matrix $P$ used above), resulting in twice the optimizer state memory required compared to RSO. GaLore projects each gradient matrix from $\mathbb{R}^{n \times n}$ to $\mathbb{R}^{r \times n}$, resulting in the same optimizer state memory requirement of $24nr$.

### A.3. Memory for Activations Analysis

From the backward propagation process, it is evident that the activations generated during forward propagation are required. Specifically, in (11), the matrices $X, Q, K, V$ need to be stored, resulting in the following memory requirement: $M_1 = 4sn$.

However, in our RSO algorithm, $X$ is only needed to compute $\mathcal{D}B_q, \mathcal{D}B_k, \mathcal{D}B_v$, where only the projections $XP_v, XP_q, XP_k$ are required. By setting $P_q = P_k = P_v = P$, we only need to store $XP$, reducing the memory requirement to $\tilde{M}_1 = 3sn + sr$.

Additionally, in (12), the matrices $\tilde{A}_s, A_s, A_h, A_o$ must be stored, requiring $M_2 = 2s^2 + 2sn$. It is worth noting that $\frac{\tilde{A}_s}{\sqrt{n}}$ does not need to be stored, as $A_s$ can be used to recover it due to the properties of the softmax function. For the RSO algorithm, storing $A_h$ and $A_o$ is unnecessary, as $A_h P_o$ and $A_o P_1$ suffice. Consequently, the memory requirement is reduced to $\tilde{M}_2 = 2s^2 + 2sr$.

For the feed-forward network in (13), the matrices $\tilde{Z}_1, Z_1, Z_2$ need to be stored, resulting in a memory requirement of $M_3 = 9sn$. In the RSO algorithm, $Z_1$ can be replaced with $Z_1 P_2$, reducing the memory requirement to $\tilde{M}_3 = 5sn + sr$.

Combining these results, the total memory cost for activations in the RSO algorithm is

$$\tilde{M}_{\text{total}} = 8sn + 2s^2 + 4sr,$$

compared to the memory cost in Adam or GaLore:

$$M_{\text{total}} = 15sn + 2s^2.$$

As the LoRA method trains parameters $A$ (corresponding to $P$ in our method), it requires the same activations as Adam, resulting in the same memory overhead as Adam or GaLore.

## B. Convergence Analysis

In this section, we present the convergence analysis of the RSO algorithm and provide a detailed proof of Theorem 5.5.

Under Assumption 5.3, the following properties hold and can be straightforwardly derived:

$$\|\boldsymbol{P}\boldsymbol{W}\| = \sqrt{\sum_\ell \text{tr}(W_\ell^\top P_\ell^T P_\ell W_\ell)} \leq \sqrt{\max_\ell \{m_\ell/r_\ell\}} \|\boldsymbol{W}\|,$$

where $\boldsymbol{W} = \{W_\ell\}_{\ell=1}^{\mathcal{L}}$ denotes any family of matrices with $W_\ell \in \mathbb{R}^{m_\ell \times r_\ell}$. For simplicity, we will not explicitly reference these properties when they are used.

**Lemma B.1.** *Under Assumptions 5.2 and 5.3, $g^k(\boldsymbol{B})$ is $(\frac{1}{\eta^k} - \hat{L})$-strongly convex and $(\frac{1}{\eta^k} + \hat{L})$-smooth, with $0 < \eta < 1/\hat{L}$ and $\hat{L} = \max_\ell \{m_\ell/r_\ell\}L$.*

*Proof.* We denote the gradient of $f$ with respect to the parameters in the $\ell$-th layer by $\nabla_\ell f(\boldsymbol{W})$. Let $h^k(\boldsymbol{B}) := f(\boldsymbol{W}^k + \boldsymbol{P}^k\boldsymbol{B})$. It can be shown that $h^k$ is $\hat{L}$-Lipschitz smooth, as follows:

$$\|\nabla h^k(\boldsymbol{B}^1) - \nabla h^k(\boldsymbol{B}^2)\|^2 = \sum_{\ell=1}^{\mathcal{L}} \left\|\left(\frac{\partial f}{\partial B_l^1}(\boldsymbol{W}^k + \boldsymbol{P}^k\boldsymbol{B}_\ell^1) - \frac{\partial f}{\partial B_\ell^2}(\boldsymbol{W}^k + \boldsymbol{P}^k\boldsymbol{B}^2)\right)\right\|_F^2$$

$$= \sum_{\ell=1}^{\mathcal{L}} \left\|(P_\ell^k)^\top (\nabla_\ell f(\boldsymbol{W}^k + \boldsymbol{P}^k\boldsymbol{B}^1) - \nabla_\ell f(\boldsymbol{W}^k + \boldsymbol{P}^k\boldsymbol{B}^2))\right\|_F^2$$

$$\leq \sum_{\ell=1}^{\mathcal{L}} \|P_\ell^k\|_F^2 \|\nabla_\ell f(\boldsymbol{W}^k + \boldsymbol{P}^k\boldsymbol{B}^1) - \nabla_\ell f(\boldsymbol{W}^k + \boldsymbol{P}^k\boldsymbol{B}^2))\|_F^2$$

$$\leq \max_\ell \left(\frac{m_\ell}{r_\ell}\right) \sum_{\ell=1}^{\mathcal{L}} \|\nabla_\ell f(\boldsymbol{W}^k + \boldsymbol{P}^k\boldsymbol{B}^1) - \nabla_\ell f(\boldsymbol{W}^k + \boldsymbol{P}^k\boldsymbol{B}^2))\|_F^2$$

$$= \max_\ell \left(\frac{m_\ell}{r_\ell}\right) \|\nabla f(\boldsymbol{W}^k + \boldsymbol{P}^k\boldsymbol{B}^1) - \nabla f(\boldsymbol{W}^k + \boldsymbol{P}^k\boldsymbol{B}^2))\|^2$$

$$\leq L^2 \max_\ell \left(\frac{m_\ell}{r_\ell}\right) \|\boldsymbol{P}^k\boldsymbol{B}^1 - \boldsymbol{P}^k\boldsymbol{B}^2\|^2 \leq \hat{L}^2\|\boldsymbol{B}^1 - \boldsymbol{B}^2\|^2.$$

As $h^k$ is $\hat{L}$-Lipschitz smooth, we can conclude that $g^k(\boldsymbol{B}) = h^k(\boldsymbol{B}) + \frac{1}{2\eta^k}\|\boldsymbol{B}\|^2$ is $(\frac{1}{\eta^k} + \hat{L})$-smooth. Furthermore, we have the inequality

$$|h^k(\boldsymbol{B}^2) - h^k(\boldsymbol{B}^1) - \langle\nabla h^k(\boldsymbol{B}^1), \boldsymbol{B}^2 - \boldsymbol{B}^1\rangle| \leq \frac{\hat{L}}{2}\|\boldsymbol{B}^1 - \boldsymbol{B}^2\|^2.$$

Based on this inequality, with the definition of $g^k$, we have

$$g^k(\boldsymbol{B}^2) \geq g^k(\boldsymbol{B}^1) - \frac{1}{2\eta^k}\|\boldsymbol{B}^1\|^2 + \frac{1}{2\eta^k}\|\boldsymbol{B}^2\|^2 + \langle\nabla h^k(\boldsymbol{B}^1), \boldsymbol{B}^2 - \boldsymbol{B}^1\rangle - \frac{\hat{L}}{2}\|\boldsymbol{B}^1 - \boldsymbol{B}^2\|^2$$

$$= g^k(\boldsymbol{B}_1) + \left\langle\nabla h^k(\boldsymbol{B}_1) + \frac{1}{\eta^k}\boldsymbol{B}_1, \boldsymbol{B}_2 - \boldsymbol{B}_1\right\rangle + \left(\frac{1}{2\eta^k} - \frac{\hat{L}}{2}\right)\|\boldsymbol{B}_1 - \boldsymbol{B}_2\|^2$$

$$= g^k(\boldsymbol{B}_1) + \langle\nabla g^k(\boldsymbol{B}_1), \boldsymbol{B}_2 - \boldsymbol{B}_1\rangle + \left(\frac{1}{2\eta^k} - \frac{\hat{L}}{2}\right)\|\boldsymbol{B}_1 - \boldsymbol{B}_2\|^2,$$

which shows that $g^k(\boldsymbol{B})$ is $(\frac{1}{\eta^k} - \hat{L})$-strongly convex when $0 < \eta < 1/\hat{L}$. $\qquad\square$

**Theorem B.2** (Theorem 5.5). *Under Assumptions 5.2 and 5.3, let the subproblem (4b) is solved from the initial point $\boldsymbol{B}^0 = \boldsymbol{0}$ to an expected $\epsilon$-inexact solution $\tilde{\boldsymbol{B}}^k$ with $\eta^k = \frac{1}{2\hat{L}}$, the sequence $\{\boldsymbol{W}^k\}$ generated by the RSO method satisfies*

$$\frac{1}{K}\sum_{k=0}^{K-1}\mathbb{E}[\|\nabla f(\boldsymbol{W}^k)\|^2] \leq \frac{18\hat{L}(f(\boldsymbol{W}^0) - f^\star)}{K} + 18\hat{L}\epsilon. \tag{19}$$

*Proof.* For simplicity, we denote $\mu := \frac{1}{\eta^k} - \hat{L}$. As $g^k(\boldsymbol{B})$ is $\mu-$strongly convex, $\forall\ \boldsymbol{B}$, we have

$$g^k(\boldsymbol{B}_\star^k) \leq g^k(\boldsymbol{B}) - \frac{\mu}{2}\|\boldsymbol{B}_\star^k - \boldsymbol{B}\|^2.$$

Let $\boldsymbol{B} = \boldsymbol{0}$ and using the definition of $\tilde{\boldsymbol{B}}^k$, we can obtain a descent condition as

$$\mathbb{E}[g^k(\tilde{\boldsymbol{B}}^k)] \leq g^k(\boldsymbol{0}) - \frac{\mu}{2}\|\boldsymbol{B}_\star^k\|^2 + \epsilon.$$

Taking the expectation with respect to the initial condition $\boldsymbol{W}^k$ and random matrix $\boldsymbol{P}^k$, by the tower rule, it can be derived

$$\mathbb{E}[g^k(\tilde{\boldsymbol{B}}^k)] \leq \mathbb{E}[g^k(\boldsymbol{0})] - \frac{\mu}{2}\mathbb{E}[\|\boldsymbol{B}_\star^k\|^2] + \epsilon.$$

Telescoping the above inequality from $k = 0$ to $K - 1$, we have

$$\sum_{k=0}^{K-1} \frac{\mu}{2} \mathbb{E}[\|\boldsymbol{B}_\star^k\|^2] \le \sum_{k=0}^{K-1} \mathbb{E}[g^k(\mathbf{0})] - \sum_{k=0}^{K-1} \mathbb{E}[g^k(\tilde{\boldsymbol{B}}^k)] + \epsilon.$$

Notice that, by the update rule of (4b), $g^{k+1}(\mathbf{0}) = f(\boldsymbol{W}^{k+1}) = f(\boldsymbol{W}^k + \boldsymbol{P}^k \tilde{\boldsymbol{B}}^k) = g^k(\tilde{\boldsymbol{B}}^k) - \frac{1}{2\eta^k}\|\tilde{\boldsymbol{B}}^k\|^2$, the terms in the RHS of the above inequality can be canceled with each other and resulting in the following inequality:

$$\sum_{k=0}^{K-1} \frac{\mu}{2} \mathbb{E}[\|\boldsymbol{B}_\star^k\|^2] \le g^0(\mathbf{0}) - \mathbb{E}[g^{K-1}(\tilde{\boldsymbol{B}}^{K-1})] - \sum_{k=0}^{K-2} \frac{1}{2\eta^k}\mathbb{E}[\|\tilde{\boldsymbol{B}}^k\|^2] + \epsilon.$$

Dividing $K$ on both sides and using the fact $g^{K-1}(\tilde{\boldsymbol{B}}^{K-1}) \ge f(\boldsymbol{W}^{K-1} + \boldsymbol{P}^{K-1}\tilde{\boldsymbol{B}}^{K-1}) \ge f^\star$, we can derive

$$\frac{1}{K}\sum_{k=0}^{K-1} \frac{\mu}{2}\mathbb{E}[\|\boldsymbol{B}_\star^k\|^2] + \frac{1}{K}\sum_{k=0}^{K-2} \frac{1}{2\eta^k}\mathbb{E}[\|\tilde{\boldsymbol{B}}^k\|^2] \le \frac{g^0(\mathbf{0}) - f^\star}{K} + \epsilon.$$

Next, we need to establish the connection between $\|\boldsymbol{B}_\star^k\|^2$ and the final stationary measure $\|\nabla f(\boldsymbol{W}^k)\|^2$. As $g^k(\boldsymbol{B})$ is $(1/\eta^k + \hat{L})$-smooth, $\forall \boldsymbol{B}$ it holds that

$$\left(\hat{L} + \frac{1}{\eta^k}\right)^2 \|\boldsymbol{B} - \boldsymbol{B}_\star^k\|^2 \ge \|\nabla g^k(\boldsymbol{B}) - \nabla g^k(\boldsymbol{B}_\star^k)\|^2 = \|\nabla g^k(\boldsymbol{B})\|^2.$$

With $\boldsymbol{B} = \mathbf{0}$, it holds

$$\frac{1}{K}\sum_{k=0}^{K-1} \frac{\mu}{2}\left(\hat{L} + \frac{1}{\eta^k}\right)^{-2}\mathbb{E}[\|\nabla g^k(\mathbf{0})\|^2] \le \frac{f(\boldsymbol{W}_0) - f^\star}{K} + \epsilon. \tag{20}$$

Furthermore, notice that $\nabla g^k(\mathbf{0}) = (\boldsymbol{P}^k)^\top \nabla f(\boldsymbol{W}^k)$ and the random matrix $\boldsymbol{P}^k$ is sampled independently to $\boldsymbol{W}^k$, for any fixed $\boldsymbol{W}^k$, we have

$$\mathbb{E}_{\boldsymbol{P}^k}[\|\nabla g^k(\mathbf{0})\|^2] = \sum_\ell \mathbb{E}_{P_\ell^k}[\mathrm{Tr}(\nabla_\ell f(\boldsymbol{W}^k)^\top P_\ell^k (P_\ell^k)^\top \nabla_\ell f(\boldsymbol{W}^k))]$$

$$= \sum_\ell \mathbb{E}_{P_\ell^k}[\mathrm{Tr}(P_\ell^k (P_\ell^k)^\top \nabla_\ell f(\boldsymbol{W}^k)\nabla_\ell f(\boldsymbol{W}^k)^\top)]$$

$$= \sum_\ell \mathrm{Tr}(\mathbb{E}_{P_\ell^k}[P_\ell^k (P_\ell^k)^\top]\nabla_\ell f(\boldsymbol{W}^k)\nabla_\ell f(\boldsymbol{W}^k)^\top)$$

$$= \sum_\ell \|\nabla_\ell f(\boldsymbol{W}^k)\|_F^2 = \|\nabla f(\boldsymbol{W}^k)\|^2.$$

Taking expectation with the respect to the randomness in $\boldsymbol{W}^k$, we can claim $\mathbb{E}[\|\nabla g^k(\mathbf{0})\|^2 = \mathbb{E}[\|\nabla f(\boldsymbol{W}^k)\|^2]$. Inserting this result back to (20) with $\eta^k = \frac{1}{2\hat{L}}$, it can be written as

$$\frac{1}{K}\sum_{k=0}^{K-1} \mathbb{E}[\|\nabla f(\boldsymbol{W}^k)\|^2] \le \frac{18\hat{L}(f(\boldsymbol{W}^0) - f^\star)}{K} + 18\hat{L}\epsilon.$$

Thus we complete the proof. □

## C. More Experimental Results

### C.1. Pre-training on LLaMA-7B Model

Table 6 compares the performance of our RSO method with GaLore and Adam on the LLaMA-7B model, where evaluations are conducted for 50K steps due to limited computational resources. We also report the memory overhead and total training time for each method. As shown in the table, RSO exhibits performance comparable to that of GaLore and Adam. Notably, RSO requires less than half the training time of the other methods.

| Method | Memory (GB) | Training Time (h) | Perplexity |
|--------|-------------|-------------------|------------|
| Adam | 78.92 | 216 | 15.43 |
| GaLore | 75.33 | 134 | 15.59 |
| **RSO** | 54.81 | 64 | 15.99 |

*Table 6.* Comparison of various pre-training methods for the LLaMA-7B model on the C4 dataset. Perplexity is reported at 50K steps. The training is conducted on $8\times$ A800 GPUs. The actual memory cost per device and the total training time are also reported. RSO and GaLore are configured with a batch size of 16, while Adam uses a batch size of 8.

## C.2. Fine-tuning on LLaMA and OPT Models

Table 7 compares RSO with other fine-tuning methods on LLaMA and OPT models across two datasets. As shown in the table, RSO outperforms other memory-efficient methods in terms of accuracy on most tasks.

| Model | WinoGrande | | | | Copa | | | |
|-------|------|------|--------|--------|------|------|--------|--------|
| | Adam | LoRA | GaLore | **RSO** | Adam | LoRA | GaLore | **RSO** |
| LLaMA-7B | 64.4 | 70.9 | 70.9 | **71.0** | 84.0 | 84.0 | 85.0 | **86.0** |
| LLaMA-13B | 73.3 | **76.6** | 74.6 | 74.7 | 90.0 | **92.0** | **92.0** | **92.0** |
| OPT-1.3B | 60.4 | 57.3 | 58.3 | **58.9** | 76.0 | 73.0 | 72.0 | **74.0** |
| OPT-6.7B | 62.2 | 64.7 | 66.8 | **69.2** | 78.0 | 80.0 | 80.0 | **82.0** |

*Table 7.* Comparison of various methods for fine-tuning LLaMA and OPT models on the WinoGrande and COPA datasets. The test accuracy for each method is reported.

| Params | Hidden | Intermediate | Heads | Layers | Steps | Data amount |
|--------|--------|--------------|-------|--------|-------|-------------|
| 60M | 512 | 1376 | 8 | 8 | 10K | 1.3 B |
| 130M | 768 | 2048 | 12 | 12 | 20K | 2.6 B |
| 350M | 1024 | 2736 | 16 | 24 | 60K | 7.8 B |
| 1 B | 2048 | 5461 | 24 | 32 | 100K | 13.1 B |
| 7 B | 4096 | 11008 | 32 | 32 | 150K | 19.7 B |

*Table 8.* Hyperparameter configurations for LLaMA models of different scales, along with the corresponding number of training steps. Due to limited computational resources, only the first 50K steps are completed for LLaMA-7B.

## D. Experimental Details

### D.1. Pre-training Experimental Setup

For the pre-training of LLaMA models across all scales, we adopt a configuration consistent with that used in (Zhao et al., 2024a). The main model hyperparameters and training steps for each method are summarized in Table 8. Specifically, in the RSO method, the Adam optimizer is used to perform multiple steps for solving each subproblem, resulting in an outer–inner iteration structure: the outer iterations correspond to subproblem updates, while the inner iterations represent Adam steps. For a fair comparison, we report the total number of inner iterations as the number of RSO steps. In all experiments, the maximum sequence length is set to 256, and the total training batch size is fixed at 512, corresponding to approximately 131K tokens per batch. A linear warm-up of the learning rate is applied over the first 10% of training steps, followed by a cosine annealing schedule that decays the learning rate to 10% of its initial value.

For the RSO method, the learning rate is selected from the set $\{0.05, 0.02, 0.01\}$. Consistent with the configuration used in GaLore, a learning rate scaling factor is applied to the weights of all multi-head attention and feed-forward layers in the model. The number of Adam steps used to solve each subproblem is set to either 200 or 500 depending on the specific experiment.

| Task | MNLI | SST-2 | MRPC | CoLA | QNLI | QQP | RTE | STS-B |
|---|---|---|---|---|---|---|---|---|
| Batch Size | 16 | 16 | 16 | 32 | 16 | 16 | 16 | 16 |
| Epochs | 30 | 30 | 30 | 30 | 30 | 30 | 30 | 30 |
| Learning Rate (Rank = 4) | 1E-05 | 3E-05 | 3E-05 | 3E-05 | 1E-05 | 1E-05 | 1E-05 | 1E-05 |
| Learning Rate (Rank = 8) | 1E-05 | 2E-05 | 2E-05 | 1E-05 | 1E-05 | 2E-05 | 2E-05 | 3E-05 |
| Scaling Factor | {8, 16, 32} | | | | | | | |
| Steps per Subproblem | {300, 500} | | | | | | | |
| Max Sequence Length | 512 | | | | | | | |

*Table 9.* Hyperparameter settings for fine-tuning the RoBERTa-Base model on the GLUE benchmark using the RSO method with different rank configurations.

## D.2. Fine-tuning Experimental Setup

**Fine-tuning on the GLUE Benchmark.** To fine-tune the pre-trained RoBERTa-Base model on the GLUE benchmark, we train for 30 epochs using a batch size of 16 across all tasks, except for CoLA, which uses a batch size of 32. Consistent with the GaLore setting, a learning rate scaling factor is applied to the weights of all multi-head attention and feed-forward layers. Detailed hyperparameter configurations are listed in Table 9.

**Fine-tuning on the WinoGrande and COPA Datasets.** For fine-tuning LLaMA and OPT models on the WinoGrande and COPA datasets, we randomly sample 1,000 training examples, 500 validation examples, and 1,000 test examples from each dataset. All experiments are run for 1,000 training steps. The corresponding hyperparameter settings are summarized in Table 10.

| Experiment | Hyperparameters | Values |
|---|---|---|
| FT | Batch Size | 16 |
| | Learning Rate | {1E-07, 1E-06, 1E-05} |
| | Weight Decay | 0 |
| LoRA | Batch Size | 16 |
| | Learning Rate | {1E-07, 1E-06, 5E-06} |
| | Rank | 8 |
| | Weight Decay | 0 |
| GaLore | Batch Size | 16 |
| | Learning Rate | {1E-07, 1E-06, 5E-06} |
| | Rank | 8 |
| | SVD Update Interval | {300, 500} |
| | Scaling Factor | {4, 8} |
| | Weight Decay | 0 |
| RSO | Batch Size | 16 |
| | Learning Rate | {1E-07, 1E-06, 5E-06} |
| | Rank | 8 |
| | Steps per Subproblem | {300, 500} |
| | Scaling Factor | {4, 8} |
| | Weight Decay | 0 |

*Table 10.* Hyperparameter settings for fine-tuning LLaMA and OPT models on the WinoGrande and COPA datasets.

