# OpenReview forum: "A Memory Efficient Randomized Subspace Optimization Method for Training Large Language Models"
_ICML.cc/2025/Conference — ICML 2025 poster_

### Official Review · Reviewer_RgSv · 2025-02-27

**Overall Recommendation:** 4

**Summary:**

The paper highlights a critical limitation in existing methods: while prior approaches have focused on reducing the memory burden of optimizer states, they have largely overlooked the substantial memory consumption imposed by activations—especially in scenarios with long context sequences or large mini-batches. To address this challenge, the authors propose a novel randomized subspace optimization framework that decomposes the high-dimensional training problem into a series of lower-dimensional subproblems. This innovative approach significantly reduces memory requirements for both activations and optimizer states while maintaining competitive performance.

**Claims And Evidence:**

The main claim of this paper is that RSO significantly reduces memory usage for optimizer states, gradients, and activations. To support this, the authors conduct a comprehensive theoretical analysis of memory consumption in Table 1, as well as empirical study in Tables 3 and Figure 2.

**Essential References Not Discussed:**

I believe that this paper have cited most related works.

**Experimental Designs Or Analyses:**

Yes. The experiments are designed to validate the memory and communication efficiency of RSO as well as its training performance. The main experiments include: 1) pretraining on LLaMA with RSO, showing that RSO could reduce memory consumption. 2) fine-tuning RoBERTa on GLUE benchmark, showing that RSO can achieve competitive performance.

**Methods And Evaluation Criteria:**

The proposed method uses a randomized subspace projection to transform the full-scale optimization problem into a series of lower-dimensional subproblems, which is well-aligned with the problem of reducing training overhead in LLMs.

**Other Comments Or Suggestions:**

No.

**Other Strengths And Weaknesses:**

Strengths:

* The paper offers a robust combination of theoretical convergence guarantees and practical memory efficiency analyses.

* Extensive experiments on both pre-training and fine-tuning verify the method’s effectiveness.

Weaknesses:

* More detailed ablation studies on the choice of the distribution of the random subspaces could offer deeper insights into the method’s sensitivity and robustness.

**Questions For Authors:**

1. For each subproblem (4a), each $\eta^k$ shoud be chosen such that $\eta^k \le 1/ \hat{L}$. How did you set this value in experiments?

2. The convergence analysis relies on specific properties of the random projection matrices. How robust is RSO when these matrices are sampled from simpler distributions (e.g., Gaussian) rather than the theoretically Haar distribution?

**Relation To Broader Scientific Literature:**

The paper is well-situated in the context of existing research on memory-efficient LLM training. It builds upon prior works such as GaLore, LoRA by addressing a key gap: the memory overhead associated with activations.

**Theoretical Claims:**

Yes, I have checked the proof of memory complexity for activations and optimizer states in Appendix A and convergence analysis in Appendix B.

---

> ### Author Rebuttal · Authors · 2025-03-31
>
> Thank you for your time and thoughtful feedback on our manuscript. Below are our detailed responses to the weaknesses and questions you raised:
>
> - Questions:
>
> 1. We would like to clarify that, as stated in our manuscript, $\eta_k$ is required to satisfy $\eta_k \le 1/\hat{L}$ to guarantee theoretical convergence. This condition is explicitly imposed to facilitate the convergence analysis. In our experiments, we set $\eta_k$ to a relatively large constant (e.g., 10) in order to accelerate convergence in practice. The results demonstrate that RSO still converges reliably under this setting.
>
> 2. In all of our experiments, we actually use Gaussian distributions instead of Haar distributions to achieve better computational efficiency. We apologize for any confusion caused by the previous version of the manuscript, and we will explicitly clarify this choice in the next revision.
>
>    As discussed in Remark 5.4, when the rank $r$ is much smaller than the embedding dimensions $m$ or $n$, Gaussian projections closely approximate the behavior of Haar projections. For completeness, we additionally provide experimental results using Haar-distributed projection matrices for training LLaMA-60M and LLaMA-130M, as shown in the following table.
>
>    | Projection Type | LLaMA-60M  | LLaMA-130M  |
>    |------------------|-------------|---------------|
>    | Gaussian         | 34.55       | 25.34         |
>    | Haar             | 34.53       | 25.06         |
>
> - Weaknesses:
>
>     As noted above, we believe this concern has already been addressed in our previous response.
>
> We hope these responses address your concerns. Please feel free to reach out with any further questions or feedback.

---

> > ### Comment · Reviewer_RgSv · 2025-04-07
> >
> > I thank the authors for the responses which have addressed my concerns. Overall, the proposed algorithm is meaningful for memory-efficient fine-tuning.

---

> > > ### Author Response · Authors · 2025-04-07
> > >
> > > Thank you for taking the time to review our manuscript and for providing valuable feedback once again.

---

### Official Review · Reviewer_75Nn · 2025-03-10

**Overall Recommendation:** 4

**Summary:**

The paper introduces a Randomized Subspace Optimization (RSO) framework for LLM training, breaking the problem into lower-dimensional subproblems to reduce memory and communication overhead. It also offers comprehensive convergence guarantees for various optimizers, with refined results for Adam.

**Claims And Evidence:**

Yes, it's great.

**Essential References Not Discussed:**

None

**Experimental Designs Or Analyses:**

The proposed RSO framework significantly improves memory efficiency and speeds up training by reducing communication overhead compared to state-of-the-art methods such as GaLore, LoRA, and Adam, while maintaining competitive performance. These results underscore the practical value of the approach.

**Methods And Evaluation Criteria:**

Yes.

**Other Comments Or Suggestions:**

See in next part.

**Other Strengths And Weaknesses:**

The theoretical analysis and experimental validations presented in the paper are robust and well-supported.

**Questions For Authors:**

[1] How do you address the subproblem in (4a)? Could you provide the detailed code implementation of how to use Adam to optimize the parameter B (how to make B get the gradient)?

**Relation To Broader Scientific Literature:**

Their work extends current literature on deep learning optimization by decomposing high-dimensional training into manageable subproblems using random subspaces. This approach not only reduces memory and communication overhead but also aligns with recent theoretical advances by providing comprehensive convergence guarantees and competitive empirical performance relative to methods like GaLore and Adam.

**Theoretical Claims:**

The paper highlights that prior methods, such as GaLore, lack comprehensive convergence guarantees—often only analyzing fixed projection matrices. In contrast, it provides a complete convergence analysis for the RSO framework across various subproblem solvers, including zeroth-, first-, and second-order methods, and optimizer variants like gradient descent and Adam.

---

> ### Author Rebuttal · Authors · 2025-03-31
>
> We appreciate your thorough review and valuable feedback on our manuscript. Below, we provide our detailed responses to the questions you raised.
>
> - Questions:
>
>    In our convergence analysis, we allow the use of different optimizers to solve each subproblem. In our experiments, we employ the Adam optimizer with a fixed number of iterations to solve each subproblem.
>
>    In our implementation of RSO, we adopt a LoRA-style design. For a linear transformation with a frozen weight matrix $W$, we introduce a trainable low-rank matrix $B$ and a projection matrix $P$. Both $W$ and $P$ are treated as fixed parameters with `requires_grad = False`, while $B$ is defined as a learnable parameter. During each forward pass, we compute $XW$ and $XPB$ separately and sum the results to obtain the final output. Since the optimizer (e.g., Adam) updates only $B$, gradients are neither computed nor stored for $W$ or $P$. As a result, PyTorch automatically avoids storing intermediate activations associated with their gradients, thereby improving memory efficiency without requiring manual intervention.
>
>    After each subproblem is solved, the update is merged into the model by setting $W \leftarrow W + P B$, followed by resampling a new random projection matrix $P$ for the next subproblem.
>
> We hope these responses address your concerns. Please feel free to reach out with any further questions or feedback.

---

### Official Review · Reviewer_FKkJ · 2025-03-14

**Overall Recommendation:** 3

**Summary:**

The paper introduces a new optimization method designed to reduce communication costs in distributed training of large language models (LLMs). The method proposed by the authors is based on breaking up the problem into smaller low-rank subproblems that we can optimize using arbitrary optimizers, by introducing a new update equation to the weights. The paper proposes optimizing over a low rank matrix passed through a random projection, which reduces the memory requirement for the activations and implicitly the gradients. The authors also provide guarantees for the number of steps required to find an approximate stationary point under mild assumptions, as well as an empirical evaluation of their method on models at various scales and across datasets, and a comparison with existing methods such as GaLore and LoRA.

**Claims And Evidence:**

The paper provides sufficient evidence that their proposed method can achieve improvements in optimization speed and memory reduction. Concretely, while the performance of RSO is similar to that of GaLore - as shown in Table 3 - for pretraining, or marginally better compared to GaLore - as shown in Table 4 - for finetuning, GaLore does not target reducing communication costs as it still requires all-gathering the whole gradient across devices. RSO on the other hand, in Table 5, shows that it can achieve much faster iteration time than Adam or GaLore during optimization.

**Essential References Not Discussed:**

None, see above.

**Experimental Designs Or Analyses:**

See methods and evaluation criteria.

**Methods And Evaluation Criteria:**

The authors sweep over hyperparameters across all baseline methods as shown in Appendix D. Due to computational constraints the authors provide training runs up to 1B, but they run fine-tuning on several datasets and larger scales than pre-training.

**Other Comments Or Suggestions:**

In general I think the paper could benefit more from a rewriting. Currently, the most important finding regarding RSO which is that even though it achieves similar performance to previous works in the literature while reducing drastically the time per iteration appears very low in the paper - Table 5. Moving this section up would greatly benefit this work.

**Other Strengths And Weaknesses:**

Strengths:
* The paper proposes a method to reduce memory usage an in turn communication cost in distributed settings
RSO achieves faster iteration time as it does not need to communicate the full gradient at each step, without sacrificing performance (when compared to the existing works such as GaLore)
* The authors perform a thorough empirical evaluation and show convergence guarantees for their method

Weaknesses:
* The method adds another layer of complexity by solving an internal suboptimization problem, which leads to additional hyperparameters to be tuned such as the optimizer choice for this problem
* The paper does not provide a FLOPs controlled experiment, see questions

**Questions For Authors:**

* When controlling for compute, how does RSO stand when compared to the other methods? Concretely, it would seemingly appear that because this method requires optimizing a subproblem for each of the K steps, this would incur a lot more FLOPs when compared to Adam/Galore etc. Could the authors provide an experiment in this direction?
* The authors mention that “when \eta_k is properly chosen” and “with suitable choice of \eta_k”, but it is not clear to me what this choice of \eta_k should be and how it should be chosen in practice. Could the authors be more specific here?
* If it is not a computational constraint, could the authors also try optimizing the subproblems with a nondiagonal preconditioner such as SOAP/Shampoo? As this requires quite a few extra runs, I will not base my decision on this experiment, but it is more of a scientific curiosity
* Why does optimizing in a random subspace given by a random rotation matrix not accumulate errors over time? More concretely, why is it the case that at each time step k we can optimize via a random projection matrix and this does not break the progress we have made so far in optimization in the previous steps?
* Could the authors provide plots of their training losses? How stable is the proposed method to hyperparameter choices? Does the learning rate change over time for the subproblem optimization?

**Relation To Broader Scientific Literature:**

The method is in general related to other low rank training works such as: [1], [2], [3], [4].

[1] uses a similar low rank approximation of the weight updates as RSO in order to minimize memory cost. [2] uses only the top subspace of the gradients in the optimization procedure, which is also a memory reduction technique. [3] uses a similar idea for the momentum term. [4] does not base its compression on rank, but on the top frequencies of the momentum term.

[1]: Hu, Edward J., et al. "Lora: Low-rank adaptation of large language models." ICLR 1.2 (2022): 3.
[2]: Zhao, Jiawei, et al. "Galore: Memory-efficient llm training by gradient low-rank projection." arXiv preprint arXiv:2403.03507 (2024).
[3]: Vyas, Nikhil, Depen Morwani, and Sham M. Kakade. "Adamem: Memory efficient momentum for adafactor." 2nd Workshop on Advancing Neural Network Training: Computational Efficiency, Scalability, and Resource Optimization (WANT@ ICML 2024). 2024.
[4]: Peng, Bowen, Jeffrey Quesnelle, and Diederik P. Kingma. "Decoupled Momentum Optimization." arXiv preprint arXiv:2411.19870 (2024).

**Theoretical Claims:**

While I have not checked the proofs of their theorems regarding guarantees, I do have several questions regarding the method and the claims that I will post in the Questions for Authors section.

---

> ### Author Rebuttal · Authors · 2025-03-31
>
> Thank you for taking the time to review our manuscript. We greatly appreciate your valuable feedback. Below, we provide our responses to your comments:
>
> - Weaknesses:
>
>    - In the RSO method, a series of subproblems need to be solved. However, this does not introduce additional complexity in tuning hyperparameters. In our experiments, for each task, we employ the Adam optimizer with the same hyperparameters across all subproblems, and apply a unified learning rate schedule rather than assigning separate schedules to individual subproblems.
>
>    - We will address the FLOPs-related discussion in the response to the questions section.
>
> - Other Comments or Suggestions:
>
>    Thank you for the suggestion. As discussed in Sections 4.3 and 6.3, RSO can also improve iteration time by reducing communication overhead when training across multiple devices, as shown in Table 5. We will move this discussion earlier in the manuscript and place greater emphasis on this point in the next revision.
>
> - Questions:
>
>    1. In our experiments, we use the Adam optimizer to perform multiple steps to solve each subproblem in RSO, resulting in an outer–inner iteration structure: outer iterations correspond to different subproblems, while inner iterations correspond to Adam steps. For fair comparison, we count the total number of inner iterations as the number of RSO steps; that is, solving one subproblem with $\tau$ Adam steps is counted as $\tau$ steps. Under this definition, the per-step computational cost of RSO is comparable to that of Adam and lower than GaLore, which incurs additional overhead due to costly SVD operations. Since all methods are evaluated under the same number of steps, we believe the computational complexity is well controlled and the comparison is fair.
>
>       In addition, RSO significantly reduces communication overhead (see Section 4.3). Table 5 compares per-step iteration time (for RSO, per inner iteration), further supporting our claim.
>
>       We apologize for any confusion and will clarify this point in the next revision.
>
>    2. The choice of $\eta_k$ is discussed in Appendix B; see Lemma B.1 and Theorem B.2 for more details, where we restrict the range of $\eta_k$ to facilitate the convergence analysis. In our experiments, we choose a relatively large constant for $\eta_k$ (e.g., 10) to accelerate convergence in practice. The results show that RSO still converges well under this setting.
>
>    3. We conduct additional experiments on LLaMA-60M, where RSO is evaluated using the Shampoo optimizer for solving each subproblem. The results are presented in the following table.
>
>       |Method|Perplexity|
>       |-|-|
>       | RSO-Adam   |34.55|
>       | RSO-Shampoo|36.45|
>
>    4. In the $k$-th outer iteration, we solve the subproblem
>    $$\min_B f(W^k + P^k B),$$
>    using a given optimizer to obtain an approximate solution $B^k$, rather than performing a single gradient step, where $P^k$ is a random projection matrix.
>
>       Since choosing $B = 0$ recovers $W^k$, it follows that $f(W^k + P^k B^k) \leq f(W^k)$. Updating the weight as $W^{k+1} = W^k + P^k B^k$ thus ensures
>       $$f(W^{k+1}) \leq f(W^k).$$
>
>       This demonstrates that solving each subproblem in a new random subspace does not compromise the progress made in previous iterations. On the contrary, each update builds upon the last, despite the randomness in the projection. As a result, errors do not accumulate across outer iterations; instead, the optimization process consistently advances in a descent direction in expectation.
>
>    5. We track both training and evaluation loss during the training of LLaMA-60M and LLaMA-130M, under configurations using 200 and 400 Adam steps per subproblem. Results are available at:
>
>       https://anonymous.4open.science/r/RSO-1CC6/
>
>       As shown, the evaluation loss decreases consistently. While the training loss may briefly fluctuate at the start of each subproblem, it also follows a downward trend overall.
>
>       We further perform an ablation study on LLaMA-60M by varying the projection rank and the number of inner Adam steps per subproblem. Results are summarized below. Increasing the rank generally improves performance, though at the cost of higher memory usage. To balance performance and efficiency, we choose rank 128—matching GaLore—for fair comparison. For the number of inner steps, moderate values (e.g., 200) perform well, whereas excessively large values may degrade performance.
>
>       |Rank|64|128|192||Inner Steps|100|200|400|800|
>       |-|-|-|-|-|-|-|-|-|-|
>       |Perplexity|37.60|34.55|33.25||Perplexity|34.54|34.55|34.80|35.30|
>
>       Finally, we adopt a unified learning rate schedule across the entire training process, rather than resetting it for each subproblem. This schedule decays adaptively, requiring no manual tuning between subproblems.
>
> We hope these responses adequately address your concerns. If you have further questions or need additional clarifications, we would be happy to provide them.

---

### Official Review · Reviewer_3u9E · 2025-03-17

**Overall Recommendation:** 4

**Summary:**

The paper introduces a memory-efficient optimization method named RSO, which optimizes subspace variables rather than the original weight. Different from the conventional subspace method such as Galore, which projects the original weight's gradient into subspace, they propose to directly optimizes the subspace variable. Hence, they can reduce the activation memory by a large margin. Such a method also avoids heavy computation of SVD as in Galore. Experiments on pre-training Llama shows that RSO achieves similar convergence speed as Galore with reduced memory and time cost.

**Claims And Evidence:**

Yes. The memory cost and convergence are carefully analyzed in this paper.

**Essential References Not Discussed:**

For the literature review of optimizer-efficient methods, it is suggested to discuss block coordinate descent methods, e.g. [1, 2], which share similar time efficiency feature as RSO by reducing gradient computational cost.

[1] BAdam: A Memory Efficient Full Parameter Optimization Method for Large Language Models

[2] BlockLLM: Memory-Efficient Adaptation of LLMs by Selecting and Optimizing the Right Coordinate Blocks

**Experimental Designs Or Analyses:**

Yes. The experimental design is fair for comparing the memory/time cost of RSO and the baseline methods. However, the RSO rank for Table 5 experiments is missing. The strategy of constructing P matrices are not explicitly discussed in the paper.

**Methods And Evaluation Criteria:**

Yes, the C4 pre-training is a standard benchmark for the considered setting.

**Other Comments Or Suggestions:**

N/A

**Other Strengths And Weaknesses:**

**Strengths**

* The proposed method reduces the memory cost of storing activation through algorithmic-level design, which is novel and may inspire development of new activation-efficient methods. Given that the activation memory is usually the main bottleneck for long-sequence training, this contribution is critical.

* Since the algorithm only computes the gradient of the subspace variable, the gradient computation is much cheaper than full backward. Hence, the time efficiency of RSO is much better than the Adam and other baseline approaches.

* The algorithm exhibits competitive performance in C4 pre-training and GLUE benchmark.

* The sample complexity result is established for RSO under strongly convex and smoothness assumption.

**Weaknesses**

* The method seems not very memory-efficient under mixed precision training setting compared to LoRA, whose master fp32 copy, gradient, and optimizer states are all low-rank matrices.  RSO cannot avoid storing the full master copy as the update will be merged into the weight after (approximately) solving each sub-problem.

**Questions For Authors:**

* What is the initialization of B matrix? The initialization may have huge influence on the convergence.

* The construction of P matrix is not explicitly discussed. In memory analysis, P is shared for query, key, and value. Is there any reason for doing this or it is mainly used for memory efficieny?

**Relation To Broader Scientific Literature:**

The key contribution of reducing the activation memory cost is of general interest for LLM research community as well as practitioners.

**Theoretical Claims:**

Yes. I went through the key proof steps of the main theorem.

---

> ### Author Rebuttal · Authors · 2025-03-31
>
> Thank you for taking the time to review our manuscript. We greatly appreciate your valuable feedback. Below, we provide our responses to your comments:
> - Experimental Designs Or Analyses:
>
>    In Table 5, we set the rank to one-fourth of the embedding dimension for both the LLaMA-1B and LLaMA-7B models, consistent with the perplexity comparison experiments and the setting used in the GaLore paper. Each matrix $P$ is constructed as a random Gaussian matrix in our experiments. We will include these implementation details in the next revision of the manuscript.
>
> - Essential References Not Discussed:
>
>    Thank you for the suggestion. These two works introduce block coordinate descent methods into LLM training. BAdam [1] divides the model parameters into several blocks and updates them sequentially, with one block optimized per epoch using Adam. BlockLLM [2] partitions the parameters by layers and updates those with larger gradient norms in each outer loop. It also incorporates a pruning mechanism to further reduce the number of parameters being updated. By limiting the number of parameters optimized in each epoch, both methods effectively reduce the memory required for storing optimizer states, as well as the computational cost associated with backpropagation.
>
>    We will incorporate these two works and discuss them in the related work section in the next revision of our manuscript.
>
> - Weaknesses:
>
>    In mixed-precision training, activations are stored in FP16, while model parameters and optimizer states are stored in FP32. Compared to LoRA, RSO incurs slightly lower memory usage for optimizer states, as LoRA trains two matrices per layer instead of one. On the other hand, RSO introduces additional memory overhead due to storing the full model in FP32. However, as illustrated in Figure 1 of the manuscript, when training large models with practical batch sizes, activation memory dominates the overall memory consumption, accounting for nearly all of the total usage. As a result, even with the extra memory required for model storage, the activation efficiency of RSO still leads to overall memory savings compared to LoRA.
>
>    Moreover, although LoRA avoids storing the full model in FP32, it significantly reduces the number of trainable parameters, which may lead to performance degradation in pre-training tasks.
>
>    We compare the actual memory consumption of RSO and LoRA during the pre-training of LLaMA-1B and LLaMA-7B under mixed-precision training. To avoid out-of-memory (OOM) errors, we set the rank-to-embedding-dimension ratio to $1/4$ for LLaMA-1B and $1/8$ for LLaMA-7B. As shown in the following table, the results confirm the memory efficiency of our RSO method even in the mixed-precision setting.
>
>    | Method | LLaMA-1B (GB) | LLaMA-7B (GB) |
>    |--------|---------------|---------------|
>    | RSO    | 54.73         | 68.39         |
>    | LoRA   | 69.93         | 78.82         |
>
> - Questions:
>
>    1. In our experiments, we use the Adam optimizer to solve each subproblem, and the matrix $B$ is initialized as a zero matrix. This ensures that the objective value of the next subproblem is initialized at the function value evaluated at the solution of the previous subproblem, which helps maintain a more stable training trajectory.
>
>    2. Each matrix $P$ is constructed as a random Gaussian matrix in our experiments. In the memory analysis, we assume a shared $P$ for the query, key, and value weight matrices to achieve optimal memory efficiency. Without sharing $P$, one would need to store $XP_q$, $XP_k$, and $XP_v$ separately instead of a single $XP$, resulting in an additional memory cost of $2sr$, which remains relatively minor.
>
> Thank you again for your valuable feedback, and we hope our responses address your concerns. If you have further questions, we would be happy to provide additional clarification.

---

> > ### Comment · Reviewer_3u9E · 2025-04-06
> >
> > Thanks for the detailed response, which resolves most of my concerns. Additionally, I missed a related work [1] during my initial review, which also reduces the memory cost of activation by utilizing low-rank structure. I suggest the authors to include more discussions on [1] in the next version of the paper.
> >
> > In summary, the proposed algorithm may serve as a competitive approach for long sequence training of relatively small models (e.g. within 8B).
> >
> > [1] LoRA-FA: Memory-efficient Low-rank Adaptation for Large Language Models Fine-tuning

---

> > > ### Author Response · Authors · 2025-04-07
> > >
> > > Thank you for the suggestion. LoRA-FA [1] modifies the original LoRA method by training only one matrix per layer while keeping the other fixed. This reduces the number of trainable parameters and also saves memory by lowering activation storage costs. However, this approach is primarily designed for fine-tuning tasks and may not be directly applicable to pre-training. We will include a discussion of this work under "activation-efficient methods" in the related work section in our next revision.

---

### Decision · Program_Chairs · 2025-05-01

**Decision:**

Accept (poster)

**Comment:**

In this paper, the authors propose a memory-efficient optimizer using random subspace optimization (RSO). Specifically, the method decomposes the original optimization problem into sub-problems via random projection matrices, and update parameters based on these projections. The authors also provide a theoretical convergence analysis for RSO. Importantly, this approach not only achieves memory efficiency through reduced optimizer states but also decreases activation memory cost due to the low-rank projections.

Reviewers all acknowledge the paper's contributions and practical significance. Furthermore, the authors' rebuttal has satisfactorily addressed most concerns raised by reviewers. Thus, this paper is recommended for acceptance.